# The Effects of a Western Diet vs. a High-Fiber Unprocessed Diet on Health Outcomes in Mice Offspring

**DOI:** 10.3390/nu15132858

**Published:** 2023-06-24

**Authors:** Elizabeth Herzl, Emily E. Schmitt, Grace Shearrer, Jill F. Keith

**Affiliations:** 1Department of Family & Consumer Sciences, University of Wyoming, Laramie, WY 82071, USAgshearre@uwyo.edu (G.S.); 2Division of Kinesiology & Health, University of Wyoming, Laramie, WY 82071, USA; eschmit4@uwyo.edu; 3WWAMI Medical Education, University of Wyoming, Laramie, WY 82071, USA

**Keywords:** fiber, Western diet, unprocessed, maternal diet

## Abstract

Diet influences critical periods of growth, including gestation and early development. We hypothesized that a maternal/early life diet reflecting unprocessed dietary components would positively affect offspring metabolic and anthropometric parameters. Using 9 C57BL-6 dams, we simulated exposure to a Western diet, a high-fiber unprocessed diet (HFUD), or a control diet. The dams consumed their respective diets (Western [*n* = 3], HFUD [*n* = 3], and control [*n* = 3]) through 3 weeks of pregnancy and 3 weeks of weaning; their offspring consumed the diet of their mother for 4.5 weeks post weaning. Measurements included dual X-ray absorptiometry (DEXA) scans, feed consumption, body weight, blood glucose, and insulin and glycated hemoglobin (HbA1c) in the offspring. Statistical analyses included one-way ANOVA with Tukey’s post hoc analysis. The offspring DEXA measures at 5 and 7.5 weeks post parturition revealed higher lean body mass development in the HFUD and control diet offspring compared to the Western diet offspring. An analysis indicated that blood glucose (*p* = 0.001) and HbA1c concentrations (*p* = 0.002) were lower among the HFUD offspring compared to the Western and control offspring. The results demonstrate that diet during gestation and early life consistent with traditional diet patterns may influence hyperglycemia and adiposity in offspring.

## 1. Introduction

Maternal nutritional status during gestation has implications on health and disease in offspring. According to the thrifty phenotype hypothesis, poor diet in utero and in early life contributes to poor growth, impaired glucose tolerance, and metabolic syndrome and can result in permanent changes to glucose metabolism [1]. Though normal metabolic shifts during pregnancy support excess glucose production for the growing fetus, blood glucose that becomes abnormally elevated by dietary intake can promote impaired fetal insulin signaling and insulin hypersecretion [2,3]. Consumption of a Western diet, characterized by processed, refined, high-glycemic, often obesogenic food patterns, can result in insulin hypersecretion and obesity [2,3,4] and has been used in studies to simulate diet-induced adverse metabolic health outcomes [5,6]. High-fat Western diets have been shown to promote weight gain and metabolic changes across generations [7,8,9,10,11,12]. Understanding the impacts of obesogenic Western diet patterns on weight gain and glucose metabolism is critical to preventing adverse metabolic changes and the development of obesity in offspring [13].

The transition from traditionally high-fiber and low-processed diets to a Western diet may underlie high rates of obesity and metabolic disease in Indigenous and minority populations [14]. Obesity is a significant and growing health concern and a known contributor to intrauterine metabolic disorders [4]. Health concerns associated with obesity, such as type 2 diabetes (T2D), gestational diabetes (GDM), and insulin resistance, are also on the rise [14,15]. The prevalence of T2D among most age groups is expected to increase globally from 415 million in 2015 to 642 million by 2040 [16].

Food processing and the impact on the quality of food, including specific sources of sugar, sodium, and trans-fats, are typically not addressed in national dietary guidance [17]. While almost all food is processed in some way before consumption, including ancient practices of drying and non-alcoholic fermentation, there are efforts to classify processed foods to enhance analysis and review in scientific assessment. The most widely used food classification system is NOVA [18,19]. According to the NOVA system, foods are classified into four groups: (1) unprocessed and minimally processed foods (e.g., edible parts of plants or animals), (2) processed culinary ingredients (e.g., oils, butter, sugar, and salt), (3) processed foods (e.g., canned or bottled foods, lunch meats, and bread), and (4) ultra-processed foods (e.g., carbonated soft drinks, candies, and instant and convenience foods) [18]. Prior to colonization and the nutrition transition to a Western diet, traditional diet patterns did not include processed or ultra-processed foods. Traditional diets kept Indigenous groups thriving and metabolic diseases low or non-existent [20,21,22]. Traditional U.S. Indigenous diets—in particular, tribal diets with ancestral homelands in the U.S. Midwest/Plains—were unprocessed, high in complex carbohydrates, fiber-rich, low in sugar, and contained ample monounsaturated and polyunsaturated fats [23,24,25,26]. Historically, the Plains tribes consumed prairie vegetation high in starch and fiber with demonstrated anti-diabetic properties [27,28,29]. Indigenous diet patterns have demonstrated preventative properties for weight gain and diabetes in multiple studies, including those demonstrating fiber’s ability to attenuate blood glucose concentrations [8,30,31,32,33]. Specifically, fiber has demonstrated regulatory effects on blood glucose by reducing the rate of nutrient absorption and T2D risk [34,35,36,37]. Eicosapentaenoic acid (EPA), an omega-3 fatty acid found in traditional Indigenous diets and one of the primary fat sources of our high-fiber unprocessed diet in the current study, has demonstrated reversed insulin resistance in C57BL/6 mice [38].

Studies in humans indicate that Indigenous diet consumption confers decreases in waist circumference, decreased cardiovascular disease risk, and greater glucose tolerance [23,29,39,40,41]. Collectively, these studies suggest that food patterns contribute to decreases in adverse metabolic outcomes (i.e., obesity, insulin resistance, and hyperglycemia) and may be central to maternal and offspring health and development. However, there is limited animal or human subject research evaluating the impacts of a pre-nutrition transition diet on maternal and infant health and biometric outcomes. We chose to use mice because mice have been used extensively in genetic investigations of human disease processes. Using mainly unprocessed or minimally processed components [18], we replicated a pre-nutrition transition diet that was high in fiber, low in sucrose, and contained ingredients consistent with dietary patterns prior to colonization. Understandably, all US tribes have unique diet patterns, so our study features common traditional components found across multiple US tribes.

The primary purpose of this study was to determine the effects of a Western diet (WD) vs. a high-fiber unprocessed (HFUD) maternal and early-life diet on offspring metabolic and anthropometric parameters. We hypothesized that offspring from dams consuming an HFUD would have lower blood glucose, insulin, and anthropometric (body weight, fat mass, and fat-free mass) measures when compared to Western diet (WD) offspring.

## 2. Materials and Methods

### 2.1. Ethical Considerations

The animal experimental protocols were approved by the University of Wyoming (UW)’s Institutional Animal Care and Use Committee (IACUC) under protocol number 20210416ES00476-02. While it is possible to obtain biochemical test values from a single experimental animal over time, that experimental design was not approved by UW’s IACUC ethics committee due to stress to the animal from repeated tail cuts to obtain blood and repeated fasting. The animals were checked, and their health was monitored daily. Our experimental design followed the ‘four Rs’ principles of reduction, refinement, replacement, and responsibility [42]. First, our team made sure for ‘Reduction’ that the experimental design was set up so that the minimal number of experimental animals were used that still ensured reliable data measurements. We ‘Refined’ our experiments to ensure that no procedures were harmful to the animals to maintain their welfare. ‘Replacement’ refers to finding another experimental model altogether to study our main objective; however, we chose mice for the proposed experiments based on their size, lifespan, laboratory availability, and previously published work in health-related fields. We have chosen to use mice because mice show similar responses as humans to diet and have been used extensively in genetic investigations of human disease processes. We also chose mice based on the fact they reproduce relatively quickly with extrapolation possible for each week of mouse gestation equivalent to about a trimester in a human. Therefore, this is an excellent model to study to gain insight into human reproduction/fetal development. Lower organisms on the phylogenetic scale are not good models for human reproduction nor are they good to use in diet studies. The final R of ‘Responsibility’ is used to promote animal welfare by improvements in the animals’ social life and development. We made sure to group-house the animals where appropriate to maintain social interactions, and we provided nestlets for enrichment.

### 2.2. Procedure

The animal experimental protocols were approved by the University of Wyoming (UW)’s Institutional Animal Care and Use Committee. We used a total of 72 mice in our model investigating gestation, weaning, and early life diet impacts on offspring metabolic parameters. Throughout the experiment, the mice were kept in a temperature-controlled environment with a 12 h dark–light cycle. Nine female mice were ordered from Jackson Lab, Bar Harbor, ME. The mice arrived at 10 weeks of age. By the 6th week of an animal’s life, mice undergo puberty. By 8 weeks, most mice are sexually mature [43]. After appropriate quarantine, we individually housed 9 female C57BL/6 mice and fed them a control diet (LabDiet 5001) for three weeks.

After 3 weeks of consuming the lab control diet, the female mice underwent a dual X-ray absorptiometry (DEXA) scan (anesthetized using isoflurane) after which they were mated with a male C57BL/6 mouse. For mating, one male mouse was housed with two females overnight or until copulation occurred. Copulation was verified by a vaginal plug and by weighing the mice daily. After pregnancy was confirmed, the dams were split into 3 groups according to their diet:(1)Group 1 dams (*n* = 3) switched from the control diet to a high-fiber, low-sugar, high-monounsaturated and -polyunsaturated-fat diet (HFUD);(2)Group 2 dams (*n* = 3) switched from the control diet to a low-fiber, high-sugar, high-saturated-fat diet (WD);(3)Group 3 dams (*n* = 3) remained on the control diet [CD].

The dams were placed in individual cages and fed their respective diets throughout pregnancy and weaning. The group 1 HFUD dams had a total of *n* = 10 female pups and *n* = 7 male pups; the group 2 WD dams had a total of *n* = 4 female pups and n = 4 male pups; and the group 3 CD dams had a total of *n* = 2 female pups and *n* = 8 male pups. Due to the lack of offspring survival after multiple pregnancies, CD dam 3 and WD dam 1 did not move to the weaning stage. In laboratory settings, mice are generally weaned in the third week after birth [44]. In addition, three weeks in a laboratory mouse is indicative of the beginning of the juvenile period of the animal and lasts about five weeks or 28–35 days [45]. Once weaned, the pups were separated into cages according to sex and maternal diet. At 8 weeks after birth, the dams and pups were sacrificed. The dams and pups were fasted overnight and subjected to euthanasia by Fatal-Plus (390 mg/mL pentobarbital sodium). Fatal-Plus was administered via intraperitoneal bolus at 100 uL per mouse [46]. The heart, liver, and skeletal muscle (gastrocnemius) were extracted and immediately frozen for future use. Blood was collected, spun down to serum, and stored in a minus 80 Celsius freezer until ELISA analysis. Figure 1 illustrates the experimental timeline for the dams and their offspring.

### 2.3. Diet Intervention

We evaluated risk factors and associated health outcomes from a whole diet pattern perspective as opposed to a single nutrient, food item, or food group [47]. To simulate food patterns characteristic of traditional Indigenous diets with minimal processing in mice, we manipulated the macronutrient allocation and chemical composition (Table 1).

We developed a customized high-fiber, low-sugar, non-dairy, organic ingredient feed to simulate unprocessed traditional U.S. Indigenous dietary patterns (HFUD) and Western (F6724) feed from Bio-serv [5]. The feed was shipped refrigerated and immediately stored in a refrigerator for three weeks before being fed to the dams and pups. The feed was placed in a stainless-steel rack feeder allowing the mice to consume their respective diet ad libitum. The HFUD and control diets were refilled bi-monthly. Food and water were weighed every 3–4 days prior to parturition; measurements were obtained by subtracting the weight of the water bottle and metal feeder from each weight measure. We calculated the average feed intake in grams by averaging the difference in the feed weight measures. The Western feed was observed to be soft, oily, and breakable. Techniques were used to prevent crumbling but were unsuccessful. Therefore, in cages where the Western feed had crumbled (1 dam, 2 pups), new food was added approximately twice every 1.5 weeks, and the feed weights were excluded from the analysis.

#### 2.3.1. High-Fiber Unprocessed Diet (HFUD)

The organic, molecularly intact ground was the base of the HFUD to minimize processing and to mimic the consumption of complex carbohydrates in traditional U.S. Indigenous diets prior to the nutrition transition. Whole ground organic corn and cellulose increased the amount of fiber in the HFUD; high dietary fiber is a defining characteristic of Indigenous dietary patterns. The Indigenous diets did not include milk or milk products, so the experimental HFUD did not contain milk [48,49]. A mix of menhaden fish oil and sunflower oil provides the fat base and egg whites provide the protein base of the HFUD. Alpha-linolenic acid, DHA, and EPA are high in fish oil and monounsaturated fatty acids are high in sunflower oil [50]. Both of these oils (as well as their nutrient properties) are characteristic of foods that would have been consumed by Indigenous people prior to the nutrition transition [39]. Although the HFUD diet contained sunflower oil, which would be categorized as a “processed culinary ingredient” or NOVA group 2, all other ingredients met the NOVA group 1 criteria for “unprocessed or minimally processed foods” [18].

#### 2.3.2. Western Diet (WD)

The Western diet reflects processed modern obesogenic eating patterns high in fat and sugar [5]. Corn starch, maltodextrin, and sucrose provide the base of the WD to simulate refined, high glycemic index simple sugars. Anhydrous milk fat and corn oil provide the fat source, and casein is the protein source for the WD. To mimic the high-sugar and high-saturated-fat intake characteristic of processed modern eating patterns, the WD has increased levels of both sucrose and fat.

#### 2.3.3. Control Diet

The control diet (Lab diet 5001) is standard rodent chow containing low levels of fiber, sugar, and fat [51,52]. Ground, conventionally grown (non-organic) corn serves as the base of the control diet and provides high levels of carbohydrates; dehulled soybean and fish meal provide high levels of protein. Porcine animal fat provides the fat source for the control diet.

### 2.4. Measures

#### 2.4.1. Body Weight and Composition

All body weight measures were conducted on the same scale throughout the duration of the experiment (Ohaus Compass CX, H-8109, Uline, Pleasant Prairie, WI, USA). To obtain the body weight measures, the mice were placed in a graduated cylinder and weighed on an industrial weighing scale. The dam weights were measured at the beginning and end of the 3 weeks of control diet intake and immediately post sacrifice. The offspring weights were measured immediately post sacrifice. For the DEXA measures, the mice were subjected to anesthesia (isoflurane), bound to a plastic sheet, and scanned; the measures were in grams and documented as ‘lean mass’ and ‘fat mass’ totals [53]. The dams underwent DEXA scans prior to copulation and prior to sacrifice. Their offspring underwent a DEXA scan at 5 and 7.5 weeks of age. For the DEXA scans, 2 male and 2 female pups from the same mother were randomly selected from their immediate sex-related siblings to participate in the scans (*n* = 4 WD males, *n* = 4 WD females, *n* = 5 HFUD males, *n* = 5 HFUD females, and n = 5 CD males, and *n* = 2 CD females); fat and lean body mass averages were taken across the three diet groups.

#### 2.4.2. Biochemical Measures

Blood glucose measures were taken immediately post sacrifice. Using blood from dissection, we measured the blood glucose concentrations in the dams and pups using a glucometer and One Touch glucose test strips [54]. After the glucose was measured, we collected the remaining blood from the pups, spun it down to serum, and froze the serum for later use in the ELISA insulin and glycated hemoglobin (HbA1c) analysis. A Crystal Chem Ultra-Sensitive Mouse Insulin ELISA kit was used to assay for the insulin levels in *n* = 9 male pups and *n* = 8 female pups [55]. We selected 5 uL serum samples at random (3 males and 3 females from each diet group) to obtain averages for each diet group. We measured the HbA1c in pups with an HbA1c Mouse Sandwich ELISA kit using the same methodology described above.

### 2.5. Statistical Analysis

Changes in body weight, fat and lean mass, blood glucose, insulin, and HbA1c data were analyzed using general linear models. Interactions between the diet groups and sex and contrasts were modeled using estimated marginal means (EMMs). EMMs model the mean of each independent variable (diet group and sex) averaged across the other factors. EMMs provide stronger analysis for unbalanced data between factors. Tukey’s post hoc tests were used to evaluate differences between the groups. *p*-values less than 0.05 were considered significant. The data were analyzed in R (v 4.1.0).

## 3. Results

In total, 7 of the *n* = 9 dams produced viable litters (*n* = 14 male pups; *n* = 11 female pups). Table 2 outlines the mean anthropometric measures and biochemical assay outcomes in the pups.

### 3.1. Food Intake

The dams consumed between 11–24 g of feed weekly (15.60 ± 5.26). The pups consumed between 8–13 g of feed weekly (11.38 ± 1.52). Variability in feed intake was observed among the dams and their offspring in the WD offspring due to the crumbly nature of the feed, so the comparative analysis was excluded.

### 3.2. Body Weight and Composition

Table 3 summarizes the differences between the diet groups’ body weight and composition. The WD offspring’s body weight was higher than that of the HFUD (mean difference = 2.08, *p* = 0.02) and CD offspring (mean difference = 1.26, *p* = 0.01). We additionally observed a significant interaction effect between diet and sex on end body weight (*p* < 0.009). Within the male offspring, the WD resulted in a higher end body weight compared to the HFUD group (b = 4.45, SE = 1.76, *p* = 0.05). Whereas in the female offspring, the WD resulted in a higher end body weight compared to the control group (b = 5.75, SE = 2.28, *p* = 0.05). The lack of consistent differences between the sexes and compared to the whole group analysis suggests a lack of power within the sex diet groups to detect a difference. Similarly, the WD offspring exhibited higher fat mass compared to the HFUD offspring (mean difference = 3.3, *p* = 0.04) and CD offspring (mean difference = 2.93, *p* = 0.03). However, we did not observe a diet-by-sex interaction related to the change in fat mass. The difference in fat mass appears to drive the change in body weight as the lean mass was not significantly different between the groups; although the WD offspring tended to lose lean body mass (WD mean lean mass change = −0.12, SD = 2.3), the HFUD (mean lean mass change = 2.6, SD = 3.24) and CD (mean lean mass change = 2.14, SD = 3.13) offspring tended to gain lean body mass. The interaction between diet and sex was also not related to the change in fat-free mass.

### 3.3. Biochemical Assay

As outlined in Table 2, the WD offspring exhibited higher blood glucose compared to the HFUD offspring (mean difference = 64.5, *p* < 0.001). However, the CD offspring had higher HbA1c values compared to both the WD (mean difference = 0.71, *p* = 0.02) and the HFUD offspring (mean difference = 0.91, *p* = 0.005) group. The CD group additionally exhibited lower insulin compared to the WD group (mean difference = −0.04, *p* = 0.03); however, no difference in insulin was observed between the WD and HFUD offspring (mean difference = 0.02, *p* > 0.05).

## 4. Discussion

Research suggests that maternal and early-life diet has a significant influence on offspring health in later life [56]. In this study, we created an animal model of diet-induced changes in body composition, insulin sensitivity, glucose, and HbA1c concentration. The offspring subjected to either a WD or an HFUD in utero and in the eight weeks following parturition displayed trends and significant differences in body mass and metabolic outcomes, including increased lean mass, lower end body weight, and lower blood glucose, insulin, and HbA1c. Other studies have demonstrated metabolic changes in offspring after being subjected to high-fat diet intake [7,10]. The offspring in our experiment who were fed a WD averaged higher levels of fasting plasma insulin and glucose and trended towards lean body mass loss, whereas the offspring consuming the HFUD had significantly lower blood glucose than the WD offspring and the lowest HbA1c concentrations and trended towards lean mass development. Our findings indicate that high-fiber unprocessed diet-induced metabolic changes in glucose occur as early as the first 8 weeks after birth.

To our knowledge, this is the first study to use a mouse model to examine the health implications of unprocessed, high-fiber, low-sugar diet intake resembling traditional U.S. Indigenous diets. A majority of rodent studies to date have experimented with isolated dietary components (e.g., high fat and high fructose) as opposed to overall dietary patterns [7,33]. Our results endorse the findings of previous studies demonstrating the preventative effects of high-fiber, low-sugar, traditional diets on elevated blood glucose concentrations [8,33].

HbA1c is a well-established measure of average glycated hemoglobin over a three-month period [57]. The high-fiber, low-sugar, low-saturated-fat composition of the HFUD prevented elevated glucose concentrations across the offspring’s lifespan, evidenced by significantly lower HbA1c and blood glucose concentrations among the HFUD offspring when compared to the other diet groups [39,58]. The HFUD offspring’s blood glucose values were considered average, [59,60] whereas the WD offspring’s blood glucose reflected values approaching pre-diabetes in other studies using fasted C57BL-6 mice [61]. The offspring in the current experiment were fasted overnight, which may have resulted in reduced biochemical assay values in all mice, especially when considering the nocturnal nature of mice’s circadian rhythms [61,62].

Studies suggest that increases in type 2 diabetes and related metabolic disorders are a direct result of changes to dietary intake, particularly decreases in unprocessed traditional foods [21,22,63,64]. Beyond diabetes prevention, traditional Indigenous diets high in fiber confer various positive health benefits, including decreases in waist circumference, decreased metabolic disorder risk, and greater glucose tolerance [39,40,41]. Specifically, fiber regulates blood glucose by increasing gut transit time, lowering postprandial blood glucose response, and slowing nutrient absorption [34,35,36,65]. Insoluble fiber, found in high amounts in our HFUD (cellulose), has demonstrated an aptitude for reducing type 2 diabetes risk [34]. Though we cannot confirm the health benefits relating to a specific nutrient or diet component, the HFUD offspring had significantly lower blood glucose values when compared to the WD offspring.

Maternal high-fat diet exposure (as with the WD in this study) in utero and during gestation can lead to permanent glucose and insulin changes that promote increased adiposity [1,11,12]. WD dams had the highest average weight and gave birth to pups that, after gestational exposure and eight weeks of WD intake, had the highest average body weight among the offspring. Future studies can focus specifically on indications for transgenerational adiposity [13]. Furthermore, higher levels of blood glucose and insulin (as seen in the WD offspring) may be indicative of insulin resistance [66,67]. Our data agree with previous reports indicating that elevated insulin levels are a consequence of Western diet consumption [2,3,4]. The Western diet in our study, which resembles the standard American diet, is highly processed and contains components—like saturated fat and sugar—that contribute to elevated blood glucose and decreased insulin sensitivity over time. We observed both elevated blood glucose concentrations and higher levels of insulin in the WD offspring than in the other diet groups at the time of sacrifice.

The CD mice in our study demonstrated the lowest levels of insulin, which may be partly explained by the high soy content of the CD; soy consumption has been shown in some studies to discourage insulin resistance in mice and humans, though this theory has been disputed [68,69]. However, the CD mice also had the highest HbA1c levels, which may be indicative of the high carbohydrate content of the CD, a lack of power, other diet components like high saturated fat content, or spontaneous glucose intolerance [58,61]. Though beyond the scope of this study, adipocyte secretions (e.g., cytokines) alter insulin signaling and beta-cell glucose sensitivity [70]. Our data suggest that the blood insulin concentrations averaged the highest among the WD offspring, which—with further study—could indicate the presence of high cytokine levels contributing to decreased insulin sensitivity. Future experiments comparing cytokine levels between the WD and the HFUD mice could provide insights into the adipocyte-related effects of unprocessed diets.

Further research is needed to confirm the generational consequences associated with WD and HFUD consumption. In our experiment, the dams consuming the WD had higher blood glucose levels than the HFUD or CD dams that were not significant when compared to the other dam diet groups, though their pups, who were exposed to this diet in utero and in adolescence, developed significantly higher mean blood glucose concentrations than the other diet groups.

The strengths of our study include the novelty of the diet intervention, the body composition measure, and the use of a genetically similar mouse model to evaluate maternal and offspring health parameters. To the best of our knowledge, this is one of the few studies to shape an experimental animal diet around eating patterns and the first to shape them around US Indigenous diet patterns. We measured body composition using a gold-standard technique; furthermore, the dams were born from an inbred strain so were genetically similar and allowed for a smaller sample size.

The limitations of our study include variability in diet exposure, a small sample population, single biochemical measurements due to ethical constraints, and variability in macronutrient quantities. With unsuccessful births, some dams in our experiment had exposure to their respective diet for a longer period than the protocol. For example, WD dam 3 and CD dam 1 were re-mated, which resulted in longer diet duration and may have limited uniformity in our measures, particularly among the dams. Our study was conducted on a small rodent sample population; thus, correlation testing between the dams and their offspring was too small to denote significance. For example, of the viable litters, only two CD females survived infancy, which reduced the total CD-offspring and CD female-only sample population. Because our glucose measures were conducted post-mortem (due to regulations on tail tip measurement in pups), the offspring were fasted for a longer period than the 4 hr fasting time found in other studies using tail tip glucose measures. After false pregnancy identification, HFUD dam 1 lived approx. 2 weeks longer than HFUD dams 2 and 3. The increased lifespan may have contributed to changes in HFUD dam 1’s weight upon sacrifice. The experimental diets had some variability between the macronutrient quantities, which may make it harder to control for the dietary source of the measured outcomes.

The study duration may be perceived as both a strength and a limitation. We observed significant results on some hypothesized parameters given a shorter study length, indicating that changes occur as early as the first eight weeks of life. Lengthening the experiment’s duration may have produced more substantial and explicit evidence of significant findings from the first eight weeks.

## 5. Conclusions

Diet influences critical periods of growth, including gestation and early development. We developed a mouse model simulating the diet pattern effects on different metabolic and anthropometric outcomes in mice. Our results demonstrate that nutritional intake during gestation and early life consistent with traditional Indigenous diet patterns may prevent hyperglycemia and the adiposity of offspring. The mice that consumed the HFUD had lower blood glucose concentrations and body weights than the WD mice. Positive trends in insulin concentrations, body weight, and lean mass accrual were also observed in the pups consuming the HFUD. Future HFUD interventions with greater duration would help further delineate significant and non-significant offspring health outcomes reflective of maternal and early life diet. The outcomes of our study help to establish the metabolic health factors associated with diet in early life and can add to the growing body of research used by clinicians to inform diet therapies for metabolic disorders.

### Recommendations for Future Work

Measuring differences in inflammatory biomarkers (e.g., cytokines and prostaglandins) between the Western diet and the HFUD may provide context for diet-induced insulin resistance and the predisposition to adiposity. To better understand insulin resistance and sensitivity, insulin measures should be obtained at several points throughout the experiment.

Future research may expand on the experiment’s duration to obtain measures across offspring life. It may also be useful to consider various hormonal analytes (leptin, GLP-1, PPY, and RBP4) in relation to diet exposure. Ideally, to inform clinical practice, this research could be extended to humans, perhaps simulating an HFUD during gestation. Because GDM and its related consequences can be treated nutritionally, future studies may observe traditional dietary pattern effects on GDM symptom control. Lastly, an animal model of traditional diet intake in dams with GDM may invoke greater insight into the transgenerational effects of the maternal diet.

## Figures and Tables

**Figure 1 nutrients-15-02858-f001:**
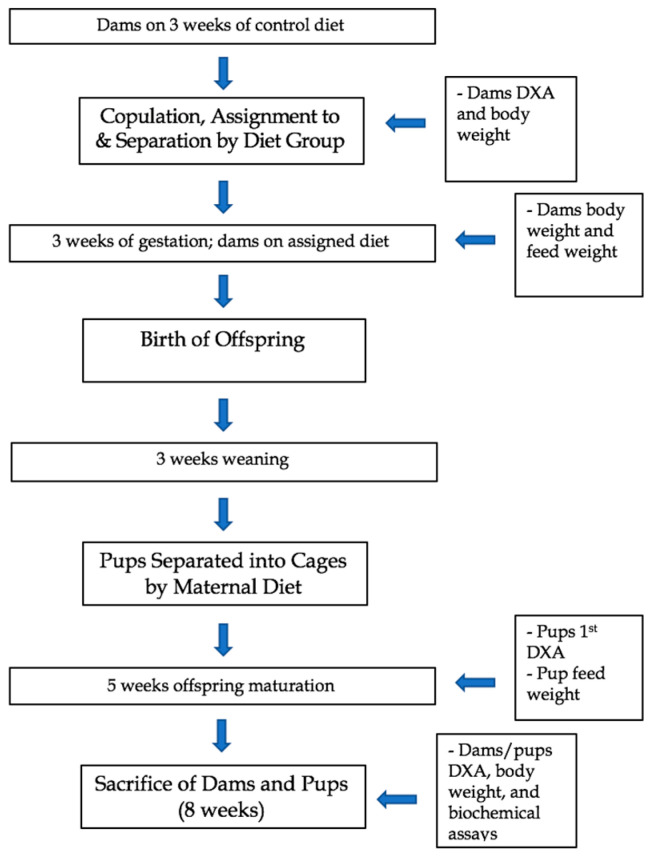
Experimental timeline and outcome measures of dams and offspring in a study examining a Western diet vs. a high-fiber unprocessed diet.

**Table 1 nutrients-15-02858-t001:** Macronutrient contribution to total kcal, sucrose, and fiber content from the three diet patterns.

	Concentration % WD	Concentration % HFUD	Concentration % CD
Protein	15.5	23	28.5
Fiber (g)	5	23.5	5.1
Carbohydrates	43.1	50	58
Fat	41.4	26	13.5
Sucrose (g)	34	0	3.7

**Table 2 nutrients-15-02858-t002:** Summary of the means by diet group.

	WD *n* = 8		HFUD *n* = 10		CD *n* = 7	
	Mean	SE	Mean	SE	Mean	SE
Final body weight	20.92	3.29	18.84	1.89	19.3	2.33
Change in fat mass	+3.5	+2.56	+0.2	+3.01	+0.57	+2.76
Change in fat-free mass	−0.12	+2.3	+2.6	+3.24	+2.14	+3.13
Final insulin *	0.12	0.03	0.1	0.02	0.08	0.02
Final HbA1c *	1.47	0.22	1.27	0.19	2.18	0.57
Final glucose	191.12	38.61	126.6	24.84	157.44	11.86

* WD *n* = 6, HFUD *n* = 6, CD *n* = 5.

**Table 3 nutrients-15-02858-t003:** Estimated marginal mean differences between the diet groups.

	WD vs. HFUD			WD vs. CD			HFUD vs. CD		
	Estimate	SE	*p*	Estimate	SE	*p*	Estimate	SE	*p*
Final body weight	2.08	0.69	0.02	2.46	0.79	0.01	0.37	0.76	0.87
Change in fat mass	3.3	1.25	0.04	4.0	1.44	0.03	0.7	1.38	0.87
Change in fat-free mass	−2.7	1.2	0.08	−3.4	1.38	0.06	−0.7	1.32	0.85
Final insulin	0.02	0.01	0.18	0.04	0.01	0.03	0.01	0.01	0.50
Final HbA1c	0.20	0.22	0.63	−0.72	0.23	0.02	−0.93	0.23	<0.01
Final glucose	64.5	13.2	<0.01	34.6	15.3	0.09	−29.9	14.6	0.13

## Data Availability

The data presented in this study are available from the corresponding author upon request.

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
