# Peer review of "The Effects of a Western Diet vs. a High-Fiber Unprocessed Diet on Health Outcomes in Mice Offspring"

_nutrients, 2023, doi:10.3390/nu15132858_

Round 1

Reviewer 1 Report

Line 79 "Primary purpose..." requires a line break

Line 80 "HUFD" is repeated and spelled out; same for WD. These two diets need to be mentioned and explained in detail, including definitions of terms, at the very beginning of the introduction.

What do we know about WD or HUFD as weaning diets (in humans) and how do they affect the development of disease in the child? And are they nutritionally appropriate as weaning foods? Is it necessary to compare these two diets on the assumption that calories, protein, carbohydrates, fat, etc. are sufficient or equivalent to begin with (different health conditions will have different physiological parameters)?

Line 85- (Materials and Methods) Please provide a separate section on ethical considerations.

Line 94- How many days after birth does 3 weeks correspond to in humans? Please mention extrapolation to humans.

Line 103- Again, abbreviations are spelled out, as is the case for CD.

Table 2.

Values that indicate a change should be marked with a sign (+/-).

Are the values for blood glucose and HbA1c averaged? Is there one measurement point per individual animal? Is it not possible to obtain biochemical test values from a single experimental animal over time in this experimental system? Does the timing of the diet have any effect?

Figures 2 and 3, 4, 5

This is a restatement of the data. It is unnecessary. Or delete the table.

It is not clear whether the purpose of this study is to examine the effects of diet during pregnancy in a mouse model or to establish the mouse model itself. If it is the latter, it should be clearly stated in the objectives. The abstract should also clearly state this. The authors state that an animal model (experimental system) has been established, but it is unclear whether this is not a new discovery because a previously reported event has been demonstrated in the present model, or whether the event observed in the mouse model is a new discovery.

Reviewer 2 Report

I was honored to review the manuscript entitled: The Effects of a Western vs. High Fiber Unprocessed Diet on Health Outcomes in Mice Offspring submitted to Nutrients. The study presents high quality and deals with important clinical issues, such type of study is needed.  I have only a few small remarks that the authors should address properly.

I recommend accepting the manuscript after minor revision.

There are only some points to correct:

 - In the “objectives” paragraph, the aim is not clearly specified, although it is understandable when reading the whole article. Could You add one clear sentence about the intention, a problem that the article is trying to solve? Maybe a hypothesis, which will be confirmed or not in the conclusion section?

 - please provide the list of abbreviations

 - please provide the number of ethical approval

  • - introduction and discussion section need improvement; please provide information on how your results will translate into clinical practice;

- in the discussion section please provide study's strong points  and study limitation section

- please correct typos

All the abovementioned issues are crucial for the credibility of the results. The paper can be accepted only after addressing all the issues and another subsequent review.

I recommend accepting the manuscript after minor revision.

Reviewer 3 Report

The authors studied the effects of a western vs. high fiber unprocessed diet (HFUD) on health outcomes in mice offspring, like body composition, glucose, insulin and HbA1c. The authors found that HFUD increased lean body mass and decreased blood glucose and HbA1c level. Furthermore, the authors also proposed the work that will be explored in the future. Overall, the data sources are compelling, and the collection of studies is well-designed to test their hypothesis. However, there are a few concerns need to be addressed about this study:

Line 99:  The samples number of dams for each group is too small. Meanwhile, how the authors confirmed the pregnancy date properly? The weight of mice won’t change a lot at the beginning of pregnancy. There is a concern that the dams may not fully exposure to their food for 21 days.

Line 208:  Please also provide the food intake figure in the manuscript.

In Figure 2 and 3, please indicate the Y axis unit. Grams? What’s the gender involved in the figure?

GTT and ITT can indicate mice glucose homeostasis. Does the author test this in this study?

Line 55: “edible parts of plans or animals”. I think the plan should be plants.

Line 17: Please provide the full name of HbA1 in the abstract.

Please reedit the results part and make it more short and more clear.

Round 2

Reviewer 1 Report

Line 283- (XX) would be appropriate to use for abbreviations. If an explanation is to be added, it would be better to change it to "WD, a high-fat diet" or something similar.

Reviewer 3 Report

N/A